# Highly Active Heterogeneous Double Metal Cyanide Catalysts for Ring-Opening Polymerization of Cyclic Monomers

**DOI:** 10.3390/polym14122507

**Published:** 2022-06-20

**Authors:** Chinh-Hoang Tran, Min-Woong Lee, Soo-Jeong Lee, Jin-Hyeok Choi, Eun-Gyeong Lee, Ha-Kyung Choi, Il Kim

**Affiliations:** School of Chemical Engineering, Pusan National University, Busandaehag-ro 63-2, Geumjeong-gu, Busan 46241, Korea; chinhtran@pusan.ac.kr (C.-H.T.); miinwoong93@pusan.ac.kr (M.-W.L.); sjlee314@pusan.ac.kr (S.-J.L.); jhchoi95@pusan.ac.kr (J.-H.C.); rud6063@pusan.ac.kr (E.-G.L.); aomghwm0116@pusan.ac.kr (H.-K.C.)

**Keywords:** double metal cyanide, epoxide, lactone, δ-valerolactone, polyols, ring-opening polymerization

## Abstract

A series of heterogeneous Zn-Co double metal cyanide (DMC) catalysts were investigated for ring-opening polymerization (ROP) of various cyclic monomers. Notably, inexpensive and commonly used organic solvents such as acetone, *N*,*N*-dimethylacetamide, *N*,*N*-dimethylformamide, dimethyl sulfoxide, nitromethane, and 1-methylpyrrolidin-2-one were very effective complexing agents for the preparation of DMC catalysts, showing high catalytic activity for the ROP of propylene oxide, ε-caprolactone, and δ-valerolactone. The chemical structures and compositions of the resultant catalysts were determined using various techniques such as FT-IR, X-ray photoelectron spectroscopy, powder X-ray diffraction, and elemental analysis. α,ω-Hydroxyl-functionalized polyether and polyester polyols with high yields and tunable molecular weights were synthesized in the presence of various initiators to control functionality. Kinetic studies of the ROP of δ-valerolactone were also performed to confirm the reaction mechanism.

## 1. Introduction

Aliphatic polyesters are an important and versatile class of biocompatible and biodegradable polymers with distinct mechanical properties that are useful in a wide range of applications from the biomedical fields such as biodegradable implant materials, drug delivery systems, and tissue engineering [1,2], to non-biomedical areas such as environmental, packaging, textile, elastomer, and polyurethane industries [3,4,5,6,7]. The development of bio-based polyesters from renewable resources has been of great interest in terms of sustainable manufacturing and environmental issues [8]. Ring-opening polymerization (ROP) of lactones is one of the most effective approaches for producing polyesters with a highly controlled manner [9]. In this context, lactide, ε-caprolactone (CL), and δ-valerolactone (VL) derived from biomasses such as lactic acid, hydroxymethylfurfural, and furfural, respectively [10,11,12], are typical candidates, and their polymerizations have been widely established [13]. The ROP of lactones was conducted using homogeneous catalytic systems such as alkoxide or phenolate complexes of Lewis acidic metals and rare-earth metals [14,15]. The metal-free, catalyzed ROP of lactones using organocatalytic systems have also gained increasing attention in recent decades [16,17,18,19,20,21,22,23]. Stannous octoate (Sn(Oct)_2_) catalyst has been used for the ROP of lactones due to the high efficiency and well-controlled polymerization [24]. However, potential toxicity induced by residual metals from the catalysts have prevented the practical use of the resultant polyester for biomedical and electronic applications [25]. Heterogeneous systems exhibit several advantages with respect to their homogeneous counterparts, notably, the ease of separation and reusability, thus diminishing the manufacturing cost and environmental issues. Nevertheless, solid catalysts for the ROP of lactones are still underexplored, as compared to homogeneous ones, thus effective catalytic systems for lactone polymerization are still in demand.

Double metal cyanide (DMC) complexes are inorganic coordination polymers generated by two metal atoms linked via cyanide bridges [26]. They are strictly Lewis acidic and highly stable solid materials [27] that have been employed as catalysts for the ROP of propylene oxide (PO) since the 1960s [28,29]. DMC catalysts have been typically prepared by reacting aqueous solutions of ZnCl_2_ and K_3_Co(CN)_6_ in the presence of *tert*-butyl alcohol (TBA) as a complexing agent (CA) and a polyvalent alcohol such as Pluronic P-123^®^ (P123) as a co-CA [30]. Thereafter, numerous organic reactions catalyzed by DMC catalysts were reported such as carbon dioxide fixation [31,32], esterification [33], transesterification [34], hydroamination [35], Prins condensation [36], oxidation reactions [37], and lactone polymerizations [38,39]. Herein, homopolymerization of various cyclic monomers including PO, CL, and VL has been investigated using various DMC catalysts in the presence of different hydroxyl-functionalized initiators (Figure 1). To improve the catalytic activity as well as to reduce the processing cost and environmental issues induced by a conventional CA such as TBA, readily available and inexpensive common solvents such as acetone (Ac), *N*,*N*-dimethylacetamide (DMAc), *N*,*N*-dimethylformamide (DMF), dimethyl sulfoxide (DMSO), nitromethane (NMe), and n-methylpyrrolidin-2-one (NMP) were used as CAs to prepare a series of DMC catalysts. The catalytic activities of these catalysts were first evaluated and optimized for the ROP of PO. Then the optimized DMC catalysts were investigated for the ROP of CL and VL. The kinetic studies of the ROP of VL were also performed to confirm the reaction mechanism.

## 2. Materials and Methods

### 2.1. Materials

Potassium hexacyanocobaltate(III) (K_3_[Co(CN)_6_]; ≥97%), anhydrous TBA (≥99.5%), poly(ethylene glycol)-b-poly(propylene glycol)-b-poly(ethylene glycol) (Pluronic^®^ P-123; molecular weight (MW) ≈ 5800), ethylene glycol (EG; ≥99%), and glycerol (GL; ≥99%) were purchased from Merck Korea (Seoul, Korea) and used without further purification. Anhydrous zinc chloride (ZnCl_2_; >98%), CL (99%), and VL (99%) were obtained from Fisher Scientific Korea Ltd. (Incheon, Korea). Polypropylene glycol (PPG; MW = 400 with functionality (F) = 2) obtained from Kumho Petrochemical Co. (Ulsan, Korea) were used as received. Reagent grades of Ac, DMAc, DMF, DMSO, NMe, NMP, toluene, chloroform, and diethyl ether were purchased from Dae Jung Chemical Co. (Gyeonggi-do, Korea) and were distilled prior to use. Polymerization grade PO, obtained from Mitsui Chemicals and SKC Polyurethanes Inc. (Ulsan, Korea), was dried over calcium hydride.

### 2.2. Preparation and Optimization of DMC Catalysts

Zn-Co DMC catalysts bearing common solvents as CAs were prepared and optimized as follows. Typically, ZnCl_2_ (1.23 g, 9 mmol), NMe (0.1–1.5 mL), and 2.5 mL of distilled water were added to a 20 mL vial (solution 1). In another vial, K_3_Co(CN)_6_ (0.5 g, 1.5 mmol) was dissolved in 2.5 mL of distilled water (solution 2). A mixture of NMe (1.0 mL) and P123 (0.1 g, 0.017 mmol) was prepared in a third vial as solution 3. Solution 1 was then reacted with solution 2 for 30 min at 50–90 °C under vigorous stirring. Subsequently, solution 3 was added to the reaction mixture and stirred for 10 min. After removal of the filtrate, the catalyst slurry was washed with excess distilled water and dried under vacuum to obtain white powder catalyst. DMC-NMe catalysts were optimized by modifying the amount of NMe in solution 1 (V = 0.1, 0.5, 1.0, and 1.5 mL) and the reaction temperature (T = 50 °C, 70 °C, and 90 °C). Likewise, DMC-DMAc, DMC-DMF, DMC-DMSO, and DMC-NMP were prepared using DMAc, DMF, DMSO, and NMP, respectively, as Cas. The pure DMC compound, DMC-1, was simply prepared by reacting aqueous ZnCl_2_ with aqueous K_3_Co(CN)_6_ at room temperature in the absence of both CAs and P123 ([Zn]/[Co] ratio of 1.5:1). DMC-TBA using TBA as a CA was prepared following the literature procedure for comparison [29].

### 2.3. Polymerization of PO

The semi-batch ROP of PO was conducted according to the previous reported protocol [40]. Typically, DMC catalyst (100 mg) and PPG-400 initiator (20 g) have been added to a 500 mL stainless steel reactor (Parr Instrument Company, Moline, IL, USA) and evacuated for 30 min to remove any trace of water. The polymerization was performed at 115 °C under continuous addition of PO monomer until the monomer consumption reach 200 g. The induction and reaction time were monitored to determine the catalytic activity.

### 2.4. Polymerization of Lactones

The batch ROP of lactone was performed in a 10 mL round-bottom flask. Prescribed amounts of the DMC catalyst, initiator, and a magnetic stirrer bar were added to the reactor and purged with nitrogen for 30 min at 90 °C. Then lactone monomers were introduced, and the polymerization was investigated under different conditions. The reaction mixture was sampled (at 30–60 min intervals) and subjected to ^1^H NMR and GPC analyses for evaluating of catalytic activity and reaction kinetics. The resultant polylactones were dissolved in chloroform, and the solid catalyst could be readily separated by centrifugation.

### 2.5. Characterization

FTIR spectra of the DMC catalysts were measured by an IR Prestige 21 spectrometer (Shimadzu Co., Tokyo, Japan) using the standard KBr disc method at a resolution of 1 cm^−1^ over a wavelength region from 4000 to 400 cm^−1^. X-ray photoelectron spectroscopies (XPS) analysis was performed by an ESCALAB 250 induced electron emission spectrometer with monochromatic Al Kα radiation (hν = 1486.6 eV) from an X-ray source operating at 12 mA and 20 kV (Thermo Fisher Scientific Solutions LLC, Seoul, Korea). The elemental analysis of the catalysts was performed using an inductively coupled plasma (ICP) optical emission spectrometer (OES) (Varian ICP720-OES) using aqua regia digestion method and a CHNS/O analyzer (Vario-Micro Cube, Elementar Analysensysteme GmbH). Thermogravimetric analysis (TGA) was conducted with a TGA Q50 analyzer (TA Instruments, New Castle, DE, USA) from 20 °C to 800 °C with a heating rate of 10 °C min^−1^. X-ray powder diffraction (XRD) analysis was performed using a RINT2000 (Rigaku Co., Tokyo, Japan) wide-angle goniometer 185 with Cu Kα radiation. The molecular weight (MW) and dispersity (Ð) were measured by gel permeation chromatography (GPC) using a Waters 150 (Agilent Technologies, Inc., Santa Clara, CA, USA) instrument operating at 40 °C with 104, 103, and 500 Å columns in tetrahydrofuran, using polystyrene standards for calibration. The hydroxyl functionality (F) of the polyols was determined by titration according to the ASTM standard (E1899-97) [41]. The ^1^H NMR (400 MHz) analysis were performed using an INOVA 400 NMR (Varian Inc., Palo Alto, CA, USA) spectrometer and referenced to the residual solvent signal of CDCl_3_ (7.25 and 77.16 ppm, respectively). The chemical shifts are presented in parts per million (ppm) relative to the residual solvent peaks as the internal standard. Matrix-assisted laser desorption ionization time-of-flight mass spectrometry (MALDI-TOF-MS) was measured by a Voyager-DETM STR Biospectrometry Workstation (Applied Biosystems Inc., Waltham, MA, USA) equipped with a nitrogen laser delivering 3 ns laser pulses at 337 nm, using dithranol as the matrix.

## 3. Results and Discussion

### 3.1. Characterization of DMC Catalysts

The structure of all the DMC catalysts were initially determined by FTIR (Figure 2). The characteristic signal of the *v*(C≡N) vibrations of the DMC catalysts bearing organic Cas had shifted to higher frequencies (2195–2190 cm^−1^), as compared to that of DMC-1 (2178 cm^−1^) bearing no organic CAs. For DMC-Ac and DMC-DMF, the characteristic signal of *v*(C=O) vibrations was located at 1701 and 1660 cm^−1^, respectively. The *δ*(Co−CN) bands of the DMC catalysts bearing organic CAs had also shifted to higher frequencies (470–474 cm^−1^), as compared to that of DMC-1 (448 cm^−1^). The signals at 1639–1615 cm^−1^ were attributed to the OH bending of the absorbed water, whereas the *v*(C−O−C) bands of P123 were observed at 1081−1089 cm^−1^ for all the catalysts, except DMC-1. Details of the FTIR vibrations are summarized in Table 1.

The XPS spectra of all the catalysts indicated that Zn, Co, O, N, and C atoms were the main components of the DMC surface (Figure 3). The binding energy of the zinc atoms in DMC-Ac, DMC-DMAc, DMC-DMF, DMC-DMSO, DMC-NMe, and DMC-NMP catalysts were slightly increased (1047.5/1024.4−1048.0/1025.1 eV), as compared to that of free ZnCl_2_ (1046.6/1023.7 eV). The binding energy of Zn 2p spin-orbital components are summarized in Appendix A.

The composition of the resultant DMC catalysts was determined by combining several elemental analysis techniques. The contents of Zn and Co (wt%) were estimated based on ICP-OES analysis, whereas CHNS/O analysis and TGA were used to determine content of the organic components. With the exception of the pure DMC compound (DMC-1), all the catalysts showed a higher content of Zn (0.48–3.37 mol per 1.0 mol of Zn_3_[Co(CN)_6_]_2_) than the stoichiometric amount of Co (1.5) due to the use of excess of ZnCl_2_. The TGA results showed that the content of the matrix water had significantly decreased from 12% for DMC-1 bearing no organic CA to 1.6–2.8% for the DMC catalysts using different organic CAs, due to the substitution of water molecules within DMC framework by CA molecules during the preparation stage (Appendix A). The percentage of CAs within the catalyst structures varied from 4.8 to 7.2%, depending on the CAs used. The estimated formulae of the resultant catalyst are summarized in Table 2.

As shown in Figure 4, the crystallinity of the DMC catalysts was significantly reduced by the substitution of the matrix water molecules for organic CAs, which was evidenced by the disappearance of the signals corresponding to the cubic phase (*Fm*-*3m*) at 2*θ* = 14.9°, 17.1°, 24.3°, 35.1°, and 39.1°. The DMC-NMe catalyst showed only a singlet peak at 23.7° 2*θ* that could be assigned to the monoclinic phase (*P11m*), whereas signals corresponding to both cubic and monoclinic phases were observed for the other DMC catalysts. These results indicated that the organic CAs had a great impact on the formation of the crystal structure and the degree of crystallinity in DMC complexes.

### 3.2. Evaluation of Catalytic Activity of DMC Catalysts for the Polymerization of PO

The DMC catalysts bearing organic solvents were briefly investigated for the ROP of PO. The optimized condition for the preparation of the DMC catalysts was determined by modifying parameters such as the amount of CA (*V*) used in solution 1 and the preparation temperature (*T*). As shown in Figure 5, DMC-DMAc prepared at 70 °C using 1 mL of CA exhibited the highest activity for the ROP of PO. Likewise, the optimized condition for making other DMC catalysts was determined. The DMC catalysts prepared in this work exhibited exceptionally high activity in the semi-batch polymerization of PO with extremely short induction times (2–11 min) and high polymerization rates, as compared to the benchmark DMC-TBA (Figure 6). The activity of the DMC catalysts for the ROP of PO (g-PPO g-cat^−1^ h^−1^) decreased in the following order: DMC-DMAc (7200) > DMC-NMe (6350) > DMC-NMP (5970) > DMC-DMF (5100) > DMC-DMSO (4480) >> DMC-TBA (2200). The resultant polyether polyols were characterized by extremely low unsaturation levels (0.0030–0.0110 meq g^−1^) and narrow dispersity (1.08–1.18) values. An overview of the catalyst preparation condition and polymerization results is given in Table 3. The ^1^H NMR spectra of the resultant PPO polyols are given in Appendix A.

### 3.3. Polymerization of CL and VL Using DMC Catalysts

Inspired by the exceptionally high activity for the ROP of PO, the performances of these catalysts were then investigated for the ROP of CL and VL. The ROP of CL was conducted at 160 °C using EG as initiator ([*CL*]_0_/[*EG*]_0_ = 10). As shown in Table 4, DMC-1 and DMC-DMSO exhibited relatively low activity (conversion of 25.9% and 38.7%, respectively), as compared to other catalysts. Among catalysts, DMC-NMP and DMC-NMe showed the highest activities for the ROP of CL with 95.3% and 95.4% monomer conversion, respectively, after 5 h of reaction time. The ^1^H NMR spectra of the PCL polyols are shown in Appendix A. After reaction, the resultant PCLs were dissolved in chloroform so that the solid catalyst could be removed by centrifugation. Leaching testing also indicates that there was no change in the monomer conversion after removal of the solid catalyst (Appendix A).

All the DMC catalysts bearing organic CAs exhibited high performances for the polymerization of VL, as calculated from ^1^H NMR spectra (Appendix A). Notably, the pure DMC compound (DMC-1) also showed high activity (81.3%), even higher than that of the conventional DMC-TBA (62.7%). It is worth noting that the DMC-1 compound showed no catalytic activity toward the ROP of PO. The activity of the DMC catalysts for the ROP of VL decreased in the following order: DMC-NMe > DMC-DMAc > DMC-DMF > DMC-NMP > DMC-DMSO > DMC-1 >> DMC-TBA. The results obtained above indicated that the crystal structure of the catalysts and the type of CAs most probably accounted for the differences in catalytic activity among the DMC catalysts. As summarized in Table 5, all the catalysts afforded PVLs with a narrow Ð of 1.16–1.26.

The polymerizations of VL were further investigated using a DMC-NMe catalyst at different [*VL*]_0_/[*EG*]_0_ and [*VL*]_0_/[*GL*]_0_ ratios. The resultant EG-PVLs and GL-PVLs exhibited high degrees of control, tunable MWs (650–6230 g mol^−1^), and low *Ð* values (1.05–1.57) (Appendix A). The GPC curves of the resultant PVLs indicated that MW increased with an increase in the [*VL*]_0_/[*I*]_0_ ratios, as expected (Appendix A). The polydispersity of the PVLs were also slightly broadened with an increase in the [*VL*]_0_/[*I*]_0_ ratios.

Figure 7 shows the ^1^H NMR spectra of the EG-PVL and GL-PVL produced by EG and GL initiators, respectively. For the EG-PVL sample, the signal corresponding to the methylene protons (a) of the EG initiator were located at 4.28 ppm while those attributed to the methylene protons of the polyester backbones appeared at 1.68, 2.34, and 4.08 ppm. The triplet signal at 3.65 ppm could be assigned to the methylene protons next to the hydroxyl end groups. For GL-PVL, the signals of the GL initiator were almost overlapped by the signals of the polymer chains. The difference in activity of the primary and secondary hydroxyl groups of the GL initiator could result in PVLs with star-like architectures, instead of linear, as obtained for the EG initiator [42].

The copolymerization of PO with CL and VL were then investigated at 140 °C using the DMC-NMe catalyst. Notably, copolymers with MWs up to 20,000 g mol^−1^ and a narrow *Ð* of 1.27 were achieved even in the absence of an initiator (Appendix A).

Kinetic studies of the ROP of VL were also conducted to corroborate the reaction mechanism (see Appendix A), and the results are summarized in Table 6. In the ROP of epoxides, the active sites of the DMC catalysts were generated by the cationic coordination between the Zn^2+^ sites and the monomers that could propagate via both the coordinative and cationic routes [43,44]. Accordingly, a coordinative pathway took place on the catalyst surface during the first stage of the polymerization with typical first-order dependence upon the monomer concentration. On the other hand, the cationic polymerization, induced by the internal Zn^2+^ sites and adsorbed monomers, occurred within the interstitial sites of the catalyst, resulting in catalyst fragmentation. This process generated new active sites for accelerating the polymerization rate and thus was one of the important factors to achieve high catalytic activity in the DMC catalysts. However, such cationic features were not observed in the DMC-catalyzed ROP of CL and VL. The kinetic results demonstrated that the polymerization of CL and VL simply proceeded via a coordination mechanism and mostly occurred on the surface of the DMC catalysts (Appendix A).

## 4. Conclusions

We have developed a feasible procedure for producing highly active heterogeneous DMC catalysts for the ROP of various cyclic monomers. Notably, the DMC catalysts were prepared using inexpensive, common solvents such as Ac, DMAc, DMF, DMSO, NMe, and NMP as alternative CAs in place of the more costly TBA. All the catalysts prepared in this work showed excellent performance for the ROPs of PO, CL, and VL, as compared to the conventional DMC-TBA. The catalytic activities for the ROP of VL were in the following order: DMC-NMe > DMC-DMAc > DMC-DMF > DMC-NMP > DMC-DMSO > DMC-1 > DMC-TBA. Subsequently, DMC-NMe were used for producing PVLs with tunable MWs (650–6230 g mol^−1^) and narrow *Ð* values (1.05–1.57) as well as PPO-PCL and PPO-PVL copolymers with MWs of 11,060 and 20,680 g mol^−1^, respectively. The polymerization kinetics were also investigated at different temperatures and various GL initiator concentrations using the DMC-NMe catalyst. The results of the kinetic studies indicated that the reaction rate was first-order dependent on the monomer and initiator concentrations, and the polymerization of VL proceeded via a coordination mechanism. The polyester polyols produced by DMC catalysts are potential precursors for the preparation of high-performance and biodegradable thermoplastic polyester and polyurethane elastomers, expanding the scope of DMC catalysis.

## Figures and Tables

**Figure 1 polymers-14-02507-f001:**
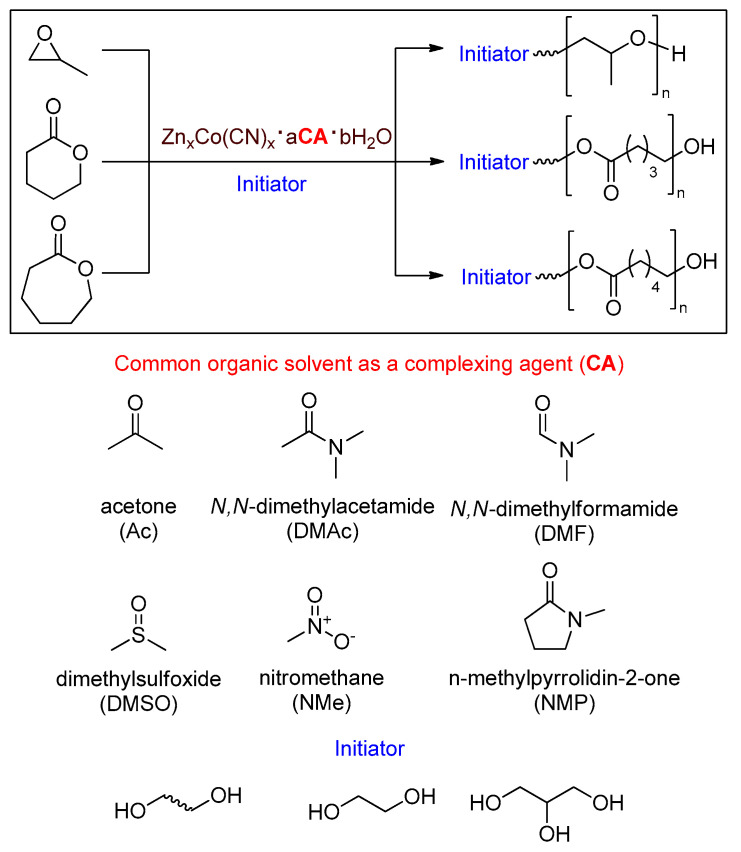
Schematic of the ROP of cyclic monomers using Zn-Co DMC catalysts prepared using various CAs.

**Figure 2 polymers-14-02507-f002:**
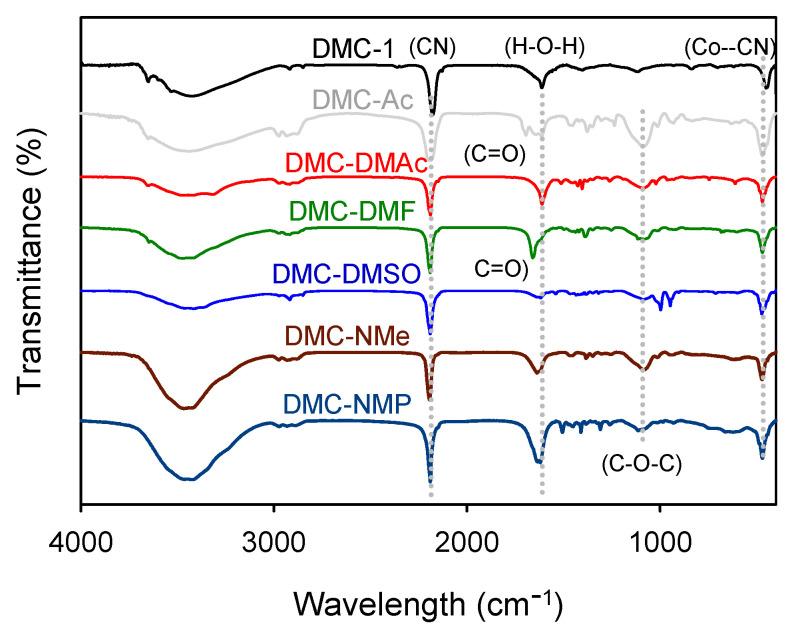
FTIR spectra of the DMC catalysts.

**Figure 3 polymers-14-02507-f003:**
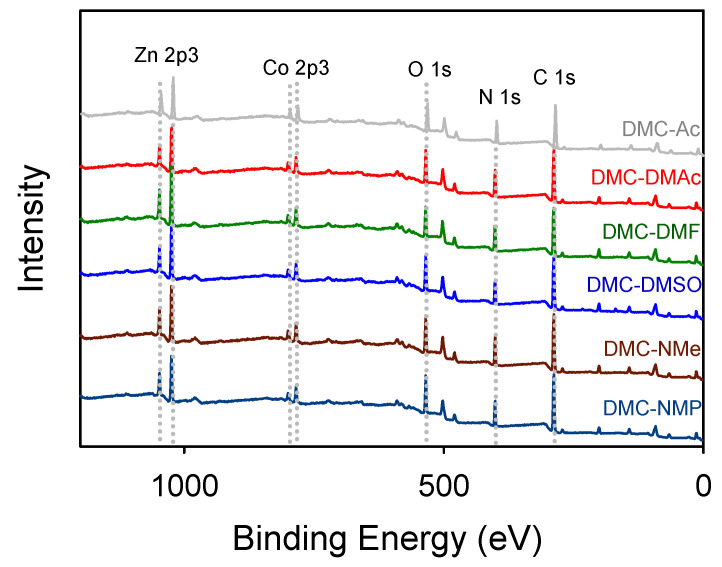
XPS spectra of the DMC catalysts.

**Figure 4 polymers-14-02507-f004:**
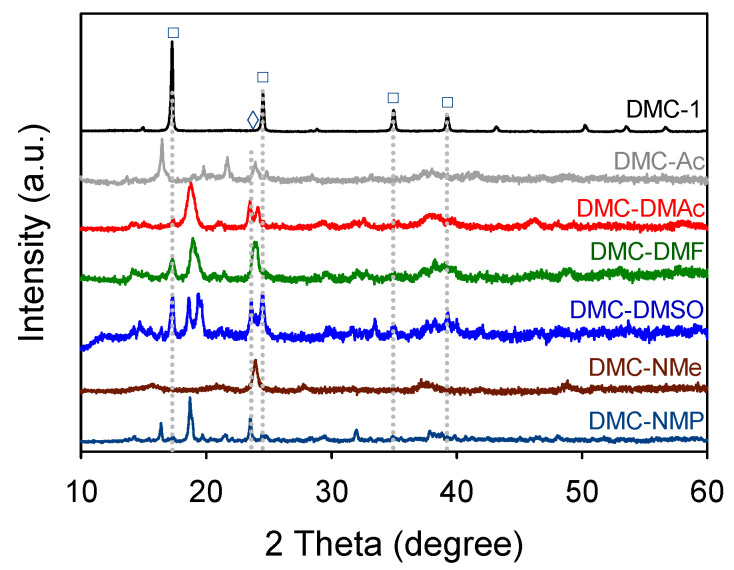
XRD patterns of the DMC catalysts. (□) and (◊) denotes the reflections corresponding to the cubic and monoclinic phases, respectively.

**Figure 5 polymers-14-02507-f005:**
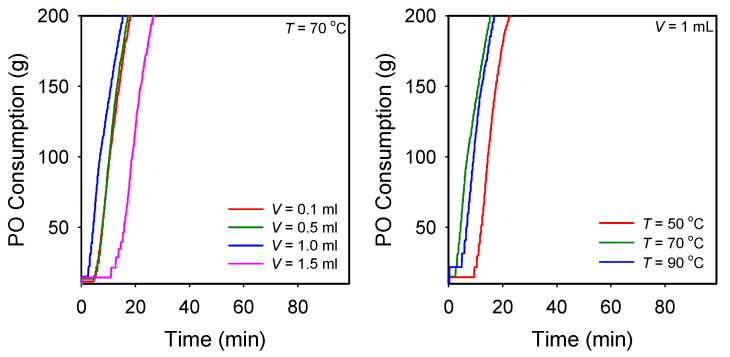
PO polymerization rate curves obtained by DMC-DMAc prepared at 70 °C using various amount of CA (left) and prepared at different temperatures using 1 mL of CA. Polymerization condition: *T*_p_ = 115 °C, PPG-400 (*F* = 2) starter = 20 g, and DMC-DMAc = 100 mg.

**Figure 6 polymers-14-02507-f006:**
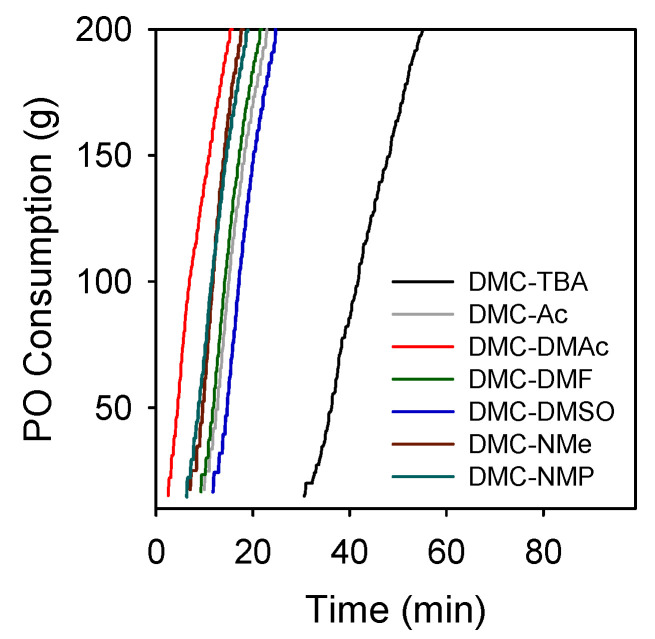
PO polymerization rate curves obtained by various DMC catalysts. Polymerization conditions: *T*_p_ = 115 °C, PPG-400 (*F* = 2) starter = 20 g, and catalyst = 100 mg.

**Figure 7 polymers-14-02507-f007:**
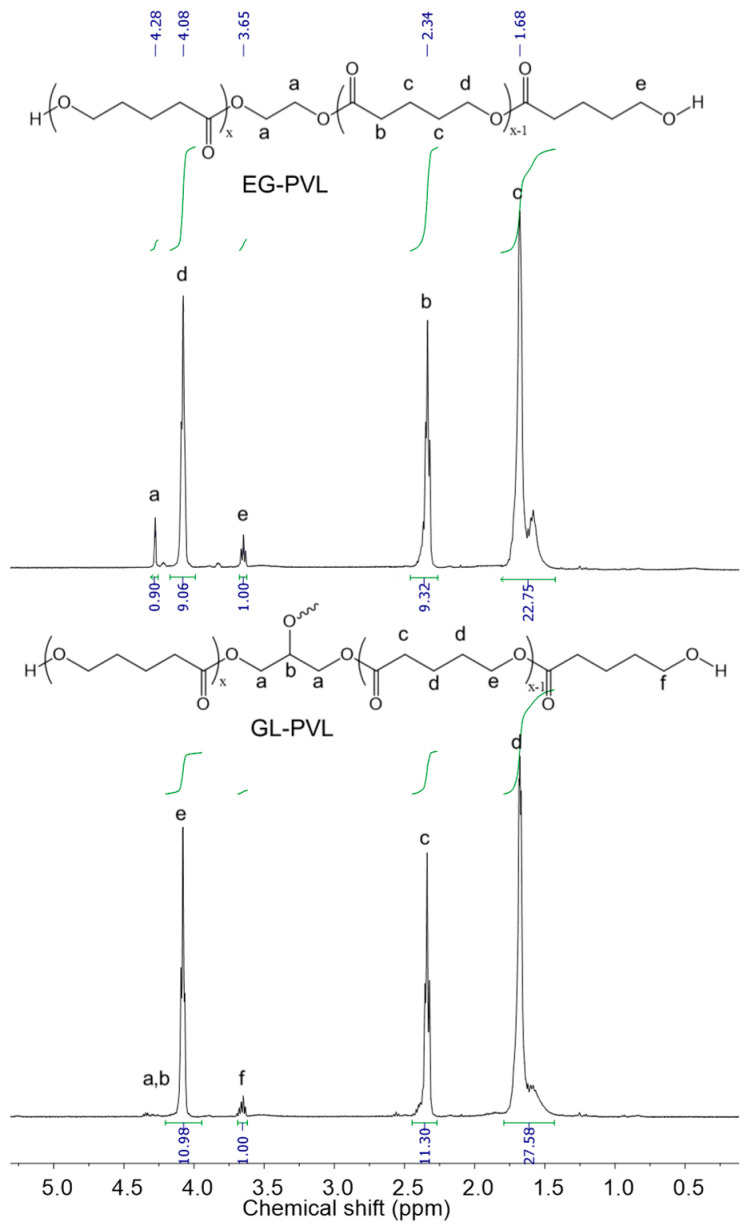
^1^H NMR spectra (400 MHz, CDCl_3_) of the EG-PVL (top) and GL-PVL (bottom) produced by DMC-NMe. Conditions: DMC-NMe = 10 mg; [*VL*] = 11 M; [*VL*]_0_/[*EG*]_0_ = 10; *T*_p_ = 160 °C.

**Table 1 polymers-14-02507-t001:** Summary of the FTIR results of DMC catalysts prepared by different CAs.

Catalyst	Vibration Frequency (cm^−1^)					
	*v* (OH)	*v*( C≡N)	*v* (C=O)	*δ* (H−O−H)	*v* (C−O−C)	*δ* (Co−CN)
DMC-1	3423	2178	−	1615	−	448
DMC-TBA	3436	2193	−	1630	1085	473
DMC-Ac	3442	2196	1701	1621	1086	471
DMC-DMAc	3417	2190	−	1616	1089	470
DMC-DMF	3471	2191	1660	1620	1081	470
DMC-DMSO	3428	2191	−	1630	1085	474
DMC-NMe	3453	2195	−	1639	1081	471
DMC-NMP	3446	2191	−	1630	1089	471

**Table 2 polymers-14-02507-t002:** Elemental analysis of the DMC catalysts.

Catalyst	ICP-Mass (wt%)		Elemental Analysis (wt%)			TGA (wt%)			Estimated Catalyst Formulation
	Zn	Co	C	H	N	CA	P123	H_2_O	
DMC-1	27.6	16.6	20.2	1.3	23.6	−	−	12.0	Zn_1.5_Co(CN)_6_·2.37H_2_O
DMC-Ac	22.3	4.13	29.2	3.5	16.1	1.7	26.8	1.0	Zn_4.87_Co(CN)_6.01_·0.32Ac·0.07P123·0.18H_2_O·13.3Cl^−^
DMC-DMAc	24.3	6.35	28.4	3.1	10.34	4.8	24.7	2.8	Zn_3.45_Co(CN)_5.99_·0.30DMAc·0.04P123·1.44H_2_O·6.56Cl^−^
DMC-DMF	27.9	11.9	29.7	3.4	19.4	4.8	20.5	2.6	Zn_2.11_Co(CN)_6.12_·0.14DMF·0.02P123·0.71H_2_O·0.78Cl^−^
DMC-DMSO	26.9	11.0	29.5	3.3	17.9	6.8	21.0	2.0	Zn_2.20_Co(CN)_6.05_·0.11DMSO·0.02P123·0.59H_2_O·1.51Cl^−^
DMC-NMe	24.9	10.7	29.3	3.2	17.4	7.2	26.9	1.6	Zn_2.10_Co(CN)_6.01_·0.65NMe·0.03P123·0.49H_2_O·0.06Cl^−^
DMC-NMP	27.3	12.4	27.1	2.9	20.2	5.8	17.2	1.8	Zn_1.98_Co(CN)_6.22_·0.28NMP·0.01P123·0.47H_2_O·1.14Cl^−^

**Table 3 polymers-14-02507-t003:** Results of the ROP of PO using various DMC catalyst.

Catalyst	Preparation Condition ^a^			Catalytic Activity		Polymer Properties		
	CA		*T* (°C)	*t*_ind_^b^ (min)	*R* _p,avg_ ^c^	GPC		Unsat. (meq g^−1^) ^d^
	Type	*V* (mL)				*M_n_*	*Ð*	
DMC-1	−	−	rt	−	−			
DMC-TBA	TBA	0.5	50	31	2200	3200	1.12	0.0065
DMC-Ac	Ac	0.1	50	11	1620			
		0.5	50	10	3880			
		1.0	50	9	4770	3900	1.10	0.01667
		1.5	50	11	4390			
DMC-DMAc	DMAc	0.1	70	4	6000			
		0.5	70	5	6320			
		1.0	70	2	7200	4200	1.09	0.0063
		1.5	70	10	4200			
		1.0	50	9	4880			
		1.0	90	4	6320			
DMC-DMF	DMF	0.1	70	8	4570			
		0.5	70	13	3430			
		1.0	70	11	3720			
		1.5	70	−	−			
		0.1	50	9	5100	4400	1.14	0.0030
		0.1	90	9	4620			
DMC-DMSO	DMSO	0.1	70	11	4480	4700	1.18	0.0091
		0.5	70	12	3680			
		1.0	70	13	2470			
		1.5	70	−	−			
		0.1	50	10	4400			
		0.1	90	17	2780			
DMC-NMe	NMe	0.1	70	−	−			
		0.5	70	8	5110			
		1.0	70	8	5430			
		1.5	70	6	6350	4500	1.14	0.0050
		1.5	50	6	6000			
		1.5	90	−	−			
DMC-NMP	NMP	0.1	70	6	5650			
		0.5	70	5	5750			
		1.0	70	23	2320			
		1.5	70	−	−			
		0.1	50	7	5370			
		0.1	90	9	5970	3800	1.08	0.0110

^a^ ZnCl2 (1.23 g, 9 mmol), K3Co(CN)6 (0.5 g, 1.5 mmol), P123 (0.3 g, 0.05 mmol). ^b^ Induction time. ^c^ Average polymerization rate in g-PPO g-cat−1 h−1. ^d^ Unsaturation level.

**Table 4 polymers-14-02507-t004:** Results of the ROP of CL using various DMC catalysts.

Catalyst	*t* (h)	Monomer Conversion (%)	NMR	GPC	
			*M_n_*	*M_n_*	*Ð*
DMC-1	5	25.9	−	−	−
DMC-TBA	5	86.7	1120	620	1.16
DMC-DMAc	5	72.7	800	500	1.20
DMC-DMF	5	80.0	900	600	1.23
DMC-DMSO	5	38.7	620	−	−
DMC-NMe	5	95.4	1170	1200	1.15
DMC-NMP	5	95.3	1190	1200	1.17

Conditions: catalyst amount ≈ 10 mg ([*Zn*]_0_ = 30 mM); [*CL*] = 9 M; [*CL*]_0_/[*EG*]_0_ = 10; *T*_p_ = 160 °C.

**Table 5 polymers-14-02507-t005:** Results of the ROP of VL using various DMC catalysts.

Catalyst	Monomer Conversion (%)		NMR	GPC	
	4 h	6 h	*M_n_*	*M_n_*	*Ð*
DMC-1	81.3	89.3	1200	1080	1.18
DMC-TBA	62.7	86.6	960	960	1.16
DMC-DMAc	87.9	89.8	1090	1250	1.25
DMC-DMF	87.2	89.1	1320	1310	1.19
DMC-DMSO	83.6	88.8	890	1150	1.26
DMC-NMe	89.6	90.3	1300	1350	1.21
DMC-NMP	85.9	90.7	1250	1230	1.24

Conditions: catalyst amount [*Zn*]_0_ = 30 mM; [*VL*] = 11 M; [*VL*]_0_/[*EG*]_0_ = 10; *T*_p_ = 160 °C.

**Table 6 polymers-14-02507-t006:** Kinetic results for the ROP of VL under various conditions using DMC-NMe and GL initiator.

*T*_p_ (°C)	[*VL*]_0_/[*GL*]_0_	Monomer Conversion (%)	*k*_app_^a^ × 10^3^ (min^−1^)	*E*_a_^b^ (kJ mol^−1^)	GPC	
					*M_n_*	*Ð*
130	10	95.3	4.12	34.66	1130	1.14
140	10	92.6	5.74		1050	1.15
150	10	94.8	7.02		1120	1.15
160	10	96.7	8.52		1150	1.15
160	5	97.1	10.10		650	1.05
160	20	95.4	6.73		2040	1.26
160	50	89.5	4.57		4600	1.41

## Data Availability

Not applicable.

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
