# Peer review of "Highly Active Heterogeneous Double Metal Cyanide Catalysts for Ring-Opening Polymerization of Cyclic Monomers"

_polymers, 2022, doi:10.3390/polym14122507_

Round 1

Reviewer 1 Report

The article 'Highly active heterogeneous double metal cyanide catalysts for ring-opening polymerization of cyclic monomers' describes the use of heterogeneous catalysts, solvated Zn-Co cyano complexes, in ROP of propylene oxide and lactones. The results are of common interest for scientists studying biodegradable polymers. The manuscript can be accepted after minor revision, taking into account the comments below.

1. In the Introduction, the authors wrote: 'potential toxicity induced by residual metals from the catalysts prevented the practical use of the resultant polyester for biomedical and electronic applications'. However, description of bulk polymerization in the Experimental does not include both separation of the polymers and trace metal analysis. Please add these data.

2. I strongly recommend to add the information about copolymerization between PO and any lactone. It would be very interesting for all readers.

Minor remarks:

Line 36 – the complexes of acidic metals used in ROP are usually alkoxides, phenolates etc., not organometallic compounds

Line 42 – the ref. [25] does not address metal toxicity

Line 75 and below – please point the manufacturers of the chemicals and equipment according to Polymers' rules (Company, City, State – if appropriate, Country)

Line 157 – in the Supporting information

Line 226 – maybe Table 5?

Line 233 – narrow MWD or low Ð value

Line 247 – integrals are needed, integration of the NMR spectra allows to determine Mn (NMR). Please calculate Mn (NMR) for PCL and PVL and add these data to Tables 4 and 5

Reviewer 2 Report

Comments for authors on manuscript polymers-1755823: "Highly active heterogeneous double metal cyanide catalysts for ring-opening polymerization of cyclic monomers” by C. H. Tran, M. W. Lee, S. J. Lee, J. H. Choi, E. G. Lee, H. K. Choi, I. Kim.

 In this manuscript the authors describe the synthesis and characterization of different double metal cyanide compounds as heterogenous catalysts for the ring-opening polymerization of different cyclic monomers such as propylene oxide (PO), ε-caprolactone (CL) and δ-valerolactone (VL). This report was carefully performed and included several interesting observations, a significant advance could find from present report. The design of new catalysts is a very attractive field for the scientific community and for the industry, therefore the manuscript is suitable for publication in Polymers after major revision:

- The authors must improve the presentation of the manuscript, 1H-NMR and 13C{1H}-NMR spectra of all polymers should be added in the supplementary material to facilitate follow-up of comments made by the authors. In the current SI, data only appears on VL polymers, PPO and PCL spectra are missing.

- The authors must correct some errors, such as; in Table 4 change [VL] = 11 by [CL] = 11; Table 1. Results of the ROP of VL using various DMC catalysts change by Table 5. ....

- The authors must make MALDI-ToF of the polymers to support the mechanistic proposal.

- The authors must explain why the catalysts are not active at moderate temperatures (rt to 90 ºC) for lactones.

Round 2

Reviewer 2 Report

Comments for authors on manuscript polymers-1755823-v2: "Highly active heterogeneous double metal cyanide catalysts for ring-opening polymerization of cyclic monomers” by C. H. Tran, M. W. Lee, S. J. Lee, J. H. Choi, E. G. Lee, H. K. Choi, I. Kim.

The manuscript has been improved by the authors and the referees’ requirements have been attended. Then, from my point of view the novelty of this manuscript is sufficient to publish in Polymers. The review is suitable for publication in Polymers.